# Training a Two-Layer ReLU Network Analytically

**DOI:** 10.3390/s23084072

**Published:** 2023-04-18

**Authors:** Adrian Barbu

**Affiliations:** Statistics Department, Florida State University, Tallahassee, FL 32306, USA; abarbu@fsu.edu

**Keywords:** neural network optimization, critical points

## Abstract

Neural networks are usually trained with different variants of gradient descent-based optimization algorithms such as the stochastic gradient descent or the Adam optimizer. Recent theoretical work states that the critical points (where the gradient of the loss is zero) of two-layer ReLU networks with the square loss are not all local minima. However, in this work, we will explore an algorithm for training two-layer neural networks with ReLU-like activation and the square loss that alternatively finds the critical points of the loss function analytically for one layer while keeping the other layer and the neuron activation pattern fixed. Experiments indicate that this simple algorithm can find deeper optima than stochastic gradient descent or the Adam optimizer, obtaining significantly smaller training loss values on four out of the five real datasets evaluated. Moreover, the method is faster than the gradient descent methods and has virtually no tuning parameters.

## 1. Introduction

Training neural networks is usually conducted using different gradient descent-based algorithms such as stochastic gradient descent or the Adam optimizer [1]. This type of training involves many passes through the entire data, usually on the order of 100–300, which makes it very slow. Moreover, the results of this training are sensitive to a number of tuning parameters such as the learning rate and the minibatch size, as well as the manner in which these parameters are changed during the training iterations (e.g., the learning rate schedule). For these reasons, many training runs are usually conducted (on the order of 10 or more) with different parameter and schedule combinations, and the most successful one is used to obtain the final model. The fact that the training takes at least 100 epochs and that, usually, at least 10 training runs are used to find a good combination implies that to train a neural network well, one must use at least 1000 passes through the data, which can be computationally expensive.

These computational reasons serve as motivation for studying a novel method for training two-layer neural networks with the square loss that needs as little as 10 passes through all the data. The method is capable of obtaining a loss that is smaller than the losses obtained by the well-tuned gradient descent algorithms and has better generalization when the number of inputs is small.

In [2], it is mentioned that “practitioners find that narrow neural nets cannot be solved well”. Our work confirms this statement experimentally, at least for neural networks (NNs) with a small number of inputs, and shows that the proposed approach can find better local minima of the loss. This work can enable a wider use of shallow NNs for many computation restricted applications. The authors of [2] also point out that the test error of an algorithm can be decomposed into the representation error, optimization error and generalization error. This work addresses the optimization error for a narrow type of NNs, namely, the two-layer NNs with ReLU-like activation and the square loss. Even though the scope is narrow, we show that in some cases, the optimization capabilities of the proposed method greatly outperform those of the standard gradient descent methods.

The contributions of this work are the following:It introduces a method for training a two-layer NN with leaky ReLU-like activation and the square loss analytically by solving ordinary least squares (OLS) equations alternatively for each layer, while keeping the activation pattern of the neurons fixed. The neuron activation pattern is a binary matrix indicating what hidden neurons fire (have non-zero activation) for each observation. It is described in Section 2.2.It conducts experiments on real and simulated data to see in what conditions the proposed method outperforms standard gradient-based optimization, such as the standard stochastic gradient descent, the Adam optimizer and the LBFGS algorithm. These experiments indicate that the proposed analytic method is faster and obtains lower loss minima when the number of observations is not very large or when the input dimension is small.

The paper is organized as follows: Section 1 gives an overview of the related work. Section 2 introduces the proposed analytic minimization method that involves fitting each layer while keeping the other layer fixed. The experimental results on real and simulated data, as well as an ablation study, are presented in Section 3. The paper finishes with conclusions and future work in Section 4.

### Related Work

Critical points of deep linear NNs (with no activation function) with the square loss have been studied in [3]. For deep linear NNs, the authors proved that every local minimum is also a global minimum, and all other critical points are saddle points. The authors have also studied the critical points of one hidden layer NNs with ReLU activation and the square loss but only in certain regions of the parameter space. They have characterized the critical points in the entire space only for an NN with one hidden node. Moreover, they have made no attempt to provide an explicit algorithm for finding the critical points. In contrast, this work presents a simple and efficient algorithm that finds the critical points for the two layers, alternatively, while keeping the neuron firing pattern fixed.

Recent work [4] has shown that training certain two-layer NNs with the square loss and stochastic gradient descent together with adequate regularization can find the global optimum of the loss function. However, these networks must have smooth and bounded activation functions, such as the sigmoid or tanh activation, and not the ReLU activation. The proof uses a Poincare-type inequality for Villani functions [5,6], which was proved in [6].

Some works [7,8] relax the NNs to be able to use convex optimization and obtain global solutions. The first work [7] is aimed at CNNs where the shared filters are represented as low rank constraints, which are relaxed for convex optimization. The second work [8] focuses on two-layer ReLU networks, and using convex duality, it shows that the two-layer ReLU network can be globally optimized with a second order cone program with a computation complexity of $O\left(d^{3}r^{3}{\left(\frac{N}{r}\right)}^{3r}\right)$, where *r* is the rank of the input data matrix (which is usually r=d for large *N*). This is polynomial time in sample size *N* but exponential in the input dimension *d*, so it can be prohibitive for large *N* and *d*. They also prove that the exponential complexity in *d* cannot be improved unless P = NP. In contrast, the algorithm introduced in this paper is faster, being $O({\left(dh\right)}^{3}+N{\left(dh\right)}^{2})$, where *h* is the number of hidden nodes, but it does not guarantee a global optimum. The convex relaxation idea was further improved in [9], where faster approximate algorithms were proposed based on gated ReLU models. However, the algorithm is gradient-based, relying on the accelerated proximal gradient method. To their advantage, the method can be applied to arbitrary loss functions, not only the square loss needed for our method.

Other works [10,11] try to develop algorithms for learning NNs in polynomial time, under the assumption that the bias terms of the hidden nodes are 0. In [10], the authors focus on two-layer NNs with ReLU activation under the assumption that the true weight matrix of the hidden nodes is full rank. They give an algorithm that is polynomial in *N* and *d* but exponential in the number of hidden nodes *h* that guarantees finding the true coefficients under certain assumptions. However, the algorithm is based on Independent Component Analysis and not on loss minimization. Again, our algorithm is cubic in the number of hidden nodes *h* at the cost of a lack of a global optimum guarantee.

The zero bias assumption is relaxed in [12], under the assumption that the weight matrix of the hidden nodes has linearly independent columns. Their algorithm is based on tensor decomposition and not on optimizing a loss function. It runs in polynomial time in *N*, *h* and *d* and recovers the true coefficients, up to a permutation and within a given error ϵ, given sufficiently many samples. However, the most important drawback is that the algorithm assumes there is no noise in the target data, so it cannot be used in practice. This kind of no noise assumption is also used in [13], where the authors show that it is possible to recover a deep ReLU NN, including its architecture, weights and biases, from the network output alone, up to an isomorphism. For that, they rely on the piecewise linear regions of the network output, which are defined by the neuron firing patterns that are also used in our work.

An argument could be made that the Broyden–Fletcher–Goldfarb–Shanno (BFGS) [14] and LBFGS (limited memory BFGS) [15] algorithms, which use an approximation of the inverse Hessian matrix, are somehow close competitors to the proposed method. However, these algorithms do not freeze the neuron firing pattern at each iteration and instead try to search for the optimum in the energy landscape of the original parameter space. The problem with these approaches is that the energy landscape has many points where the inverse Hessian is infinite because of the non-differentiable nature of the piecewise linear ReLU activation. For this reason, the BFGS and LBFGS algorithms explode near the local optima, as can be easily observed experimentally. In contrast, by freezing the neuron firing pattern in our work, the analytic solution was observed experimentally to always exist for some data (three of the five real datasets from experiments) and usually exists for the rest of the data.

Table 1 presents an overview of the above mentioned works, with their computation complexities, types of loss function they optimize and whether they provide global optimum guarantees or not.

## 2. Materials and Methods

This work will focus on two-layer ReLU networks:$$f\left(x\right)=\sigma \left(\left(1,x^{T}\right)A\right)B+b^{0}\in R^{c}$$
where $x\in R^{d}$ is an input vector, $A=(a_{1},\ldots ,a_{h})$ is the (d+1)×h matrix of weights for the hidden layer and $B=(b_{1},\ldots ,b_{c})$ is the h×c matrix of weights of the output layer and $b^{0}=\left(b_{1}^{0},\ldots ,b_{c}^{0}\right)$ are the biases for the *c* outputs. The activation functions of interest are the leaky ReLU type σ(x)=αx+(1−α)max(0,x), with α∈[0,1). In the experiments, we will look at the ReLU (α=0) and the leaky ReLU (α=0.1) (Figure 1).

Given observations $\left(x_{1},y_{1}\right),...,\left(x_{N},y_{N}\right)\in R^{d}\times R^{c}$, the square loss function for these networks is:(1)$$L\left(A,B,b^{0}\right)=\frac{1}{N}\sum_{k=1}^{c}\sum_{i=1}^{N}∥\sigma \left(\left(1,x_{i}^{T}\right)A\right)b_{k}+b_{k}^{0}-y_{ik}{∥}^{2}+\lambda (∥b^{0}{∥}_{2}^{2}+{∥B∥}_{F}^{2}+{∥A∥}_{F}^{2})),$$
where ${∥A∥}_{F}^{2}=\sum_{i,j}A_{ij}^{2}$ is the Frobenius norm.

We will use the standard notation denoting by X the matrix of observations
(2)$$X=\left(\begin{array}{cccc} 1 & x_{11} & \ldots & x_{1d}\\ & & \ldots & \\ 1 & x_{N1} & \ldots & x_{Nd} \end{array}\right),$$
and by Y the matrix of outputs, with $y_{i}^{T}$ as rows and $y_{\cdot j}$ as columns,
(3)$$Y=\left(\begin{array}{c} y_{1}^{T}\\ \ldots \\ y_{N}^{T} \end{array}\right)=\left(\begin{array}{ccc} y_{11} & \ldots & y_{1c}\\ \; & \ldots \\ y_{N1} & \ldots & y_{Nc} \end{array}\right)=\left(y_{\cdot 1},\ldots ,y_{\cdot c}\right),$$
where $y_{\cdot j}$ is the vector of targets for the *j*-th output.

We will use an algorithm that minimizes the loss (Equation 1) by alternately finding the critical points of the loss in terms of B and A, where for the critical points with respect to A, the neuron firing pattern is kept frozen. The procedure is described in Algorithm 1.
**Algorithm 1** Analytic Minimization (ANMIN) **Input:** Training data X,Y. **Output:** Trained network parameter vectors $\left(\hat{A},\hat{B},{\hat{b}}_{0}\right)$.
1:Initialize A randomly2:Fit $\left(B,b^{0}\right)$ using Equation (Equation 4).3:Compute loss $l_{0}=L\left(A,B,b^{0}\right)$4:**for** i=1 to *E* **do**5:   Update F=I(XA>0) and G=(1−α)F+α16:   Compute M using Equation (Equation 7) and c using Equation (Equation 8)7:   **if** $\ln|M+\lambda I_{(d+1)h}|>\tau$ **then**8:        Solve $(M+\lambda I_{(d+1)h})a=c$9:   **else**10:     Obtain U,D,V by SVD such that     $UDV^{T}=M+\lambda I_{(d+1)h}$11:     If $D_{ii}<0.0001$, set $D_{ii}=0.0001,i=1,\ldots ,dh$12:     Obtain $a=UD^{-1}V^{T}c$13:   **end if**14:   Reshape vector a into (d+1)×h matrix A15:   Fit $\left(B,b^{0}\right)$ using Equation (Equation 4).16:   Compute loss $l_{i}=L\left(A,B,b^{0}\right)$17:   **if** $l_{i}<l_{0}$ **then**18:      Set $\hat{A}=A,\hat{B}=B,{\hat{b}}^{0}=b^{0}$19:      Set $l_{0}=l_{i}$20:   **end if**21:**end for**


The threshold τ controls when the linear system $(M+\lambda I_{(d+1)h})a=C$ can be solved numerically. In practice, we take τ=−10,000.

### 2.1. Fitting the Output Layer Weights B

Fitting the output layer weight matrix B is simple for OLS, using $S=(\sigma \left(XA\right),1_{N})$ as the input. Since the square loss is a sum of the square losses over the output variables, B can be solved separately for each output variable:(4)$$\left(\frac{1}{N}S^{t}S+\lambda I_{h+1}\right)\left(\begin{array}{c} b_{k}\\ b_{k}^{0} \end{array}\right)=S^{T}y_{\cdot k},k=1,\ldots ,c.$$

### 2.2. Fitting the Hidden Layer Weights A

Let F=I(XA>0) be the N×h binary firing pattern matrix, indicating what hidden neurons fire for each observation. Here, I(x) is the elementwise indicator operator. Let G=(1−α)F+α1, where 1 is the matrix with all entries 1.

Then, the neural network response for training observation $\left(x_{i},y_{i}\right)$, is
$${\hat{y}}_{ik}\;=\;b_{k}^{0}+\sum_{j=1}^{h}b_{jk}G_{ij}{\tilde{x}}_{i}a_{j}\;=\;\left(G_{i}*\left({\tilde{x}}_{i}A\right)\right)b_{k}+b_{k}^{0},k\;=\;1,\ldots ,c$$
where $G_{i}$ is the *i*-th row of G, “∗” is the elementwise multiplication and ${\tilde{x}}_{i}=\left(1,x_{i}^{T}\right)$ is the *i*-th row of X. Then, all responses can be written as
$$\hat{Y}=\left(G*\left(XA\right)\right)B+b^{0}$$

Adding the firing pattern G to the loss function parameters gives
(5)$$L\left(A,B,b_{0},G\right)=\frac{1}{N}\sum_{k=1}^{c}\sum_{i=1}^{N}∥\sum_{j=1}^{h}b_{jk}G_{ij}{\tilde{x}}_{i}a_{j}+b_{k}^{0}-y_{ik}{∥}^{2}+\lambda (∥b^{0}{∥}^{2}+{∥B∥}_{F}^{2}+{∥A∥}_{F}^{2})).$$

Let
(6)$$U_{jl}=\frac{1}{N}\sum_{i=1}^{N}G_{ij}G_{il}{\tilde{x}}_{i}{\tilde{x}}_{i}^{T},$$
which are (d+1)×(d+1) matrices that can be computed incrementally using batches.

Let
(7)$$M=\left(\begin{array}{cccc} M_{11} & M_{12} & \ldots & M_{1h}\\ \ldots & \ldots & \ldots & \ldots \\ M_{h1} & M_{h2} & \ldots & M_{hh} \end{array}\right)$$
be the (d+1)h×(d+1)h matrix with the cell $M_{jl}=\sum_{k=1}^{c}b_{jk}b_{lk}U_{jl}$.

Additionally, let
(8)$$c={\left(v_{1}^{T},\ldots ,v_{h}^{T}\right)}^{T},v_{j}=\frac{1}{N}\sum_{k=1}^{c}\sum_{i=1}^{N}b_{jk}\left(y_{i}-b_{k}^{0}\right)G_{ij}x_{i}.$$

The following Theorem describes the linear system of equations that need to be solved in order to find the critical points of the loss function $L(A,B,b^{0},G)$ with respect to A analytically.

**Theorem** **1.**
*If the matrix G from Equation *(Equation 5)* is fixed, the critical points with respect to A of the loss function $L(A,B,b^{0},G)$ from Equation *(Equation 5)* are solutions of the equation:*

(9)
$$(M+\lambda I_{(d+1)h})a=c,$$

*where $a={\left(a_{1}^{T},\ldots ,a_{h}^{T}\right)}^{T}$ is the matrix A unraveled.*


**Proof.** The loss (Equation 5) can also be written as
(10)$$\begin{array}{cc} L(A,B,b_{0},G)= & \frac{1}{N}\sum_{k=1}^{c}\sum_{j,l=1}^{h}a_{j}^{T}b_{jk}b_{lk}\left(\sum_{i=1}^{N}G_{ij}G_{il}x_{i}x_{i}^{T}\right)a_{l}+\frac{2}{N}\sum_{j=1}^{h}\sum_{k=1}^{c}\sum_{i=1}^{N}b_{jk}\left(b_{k}^{0}-y_{i}\right)G_{ij}x_{i}^{T}a_{j}\\ \; & +\lambda (b^{0T}b^{0}+\sum_{k=1}^{c}b_{j}^{T}b_{k}+a^{T}a); \end{array}$$
therefore,
(11)$$L\left(A,B,b_{0},G\right)=\;\sum_{j,l=1}^{h}\;a_{j}^{T}\;\left(\sum_{k=1}^{c}b_{jk}b_{lk}U_{jl}\right)a_{l}\;-\;2\;\sum_{j=1}^{h}\;v_{j}^{T}a_{j}+\lambda \left(b^{0T}b^{0}+\sum_{k=1}^{c}b_{j}^{T}b_{k}+a^{T}a\right).$$Then, the loss function can be written using Equation (Equation 7) as
(12)$$L\left(a,\;B,\;b_{0},\;G\right)\;=\;a^{T}Ma-2c^{T}a+\lambda \left(b^{0T}b^{0}\;+\;\sum_{k=1}^{c}\;b_{j}^{T}b_{k}+a^{T}\;a\right).$$This is a quadratic function in a, and by setting its gradient with respect to a to zero, we obtain Equation (Equation 9). □

### 2.3. Computation Complexity

The computation complexity of the analytic minimization algorithm ANMIN described in Algorithm 1 can be easily calculated as follows.

Computing the input S for solving B is O(Ndh) and fitting B is $O(h^{3}+ch^{2})$. Computing each matrix $U_{ij}$ is $O(Nd^{2})$, so computing M is $O(N{\left(dh\right)}^{2})$. Fitting A is $O({\left(dh\right)}^{3}+c{\left(dh\right)}^{2})$. Since there are a fixed number of iterations, the whole algorithm is $O({\left(dh\right)}^{3}+\left(N+c\right){\left(dh\right)}^{2})$, so it is linear in *N* and *c* but cubic in *d* and *h*.

### 2.4. Datasets Used

Real data experiments will be conducted on three datasets from the UC Irvine Machine Learning repository [16] and two datasets that were specially generated for two computer vision tasks. These five datasets are summarized in Table 2.

From the UC Irvine Machine Learning repository [16] were used the abalone, bike-sharing and year prediction MSD datasets, all three being regression datasets.

The abalone dataset is about predicting the age of abalone from its physical measurements. It has 4177 observations and 7 features, so it is a low dimensional dataset.

The bike-sharing dataset is about predicting the hourly and daily bike rental counts in a bike rental system. The hourly count was used, containing 17,379 observations and 14 features. The feature that counts the number of registered users was removed since it is very strongly correlated with the response, and the remaining 13 features were used for prediction.

The year Prediction MSD dataset is about predicting the release year of a song from audio features extracted from the song. It contains 515,345 observations and 90 features.

Another dataset was generated based on the single-shape deep SDF task from [17]. It is about predicting the value of a signed distance transform (SDF) matrix from the pixel coordinates (x,y). The input shape used was the first mask image of the Weizmann horse dataset [18], shown in Figure 2, left, from which the SDF was computed. Taking all pixels’ coordinates of the image as inputs and their corresponding SDF values as targets, a dataset with 73,080 observations and two features was obtained.

All these regression datasets are about predicting a single output. To evaluate the performance of ANMIN for predicting multiple outputs, the NN was used as a denoising autoencoder (DAE). In this case, the desired outputs consist of natural image patches, and the inputs are the same patches corrupted by noise. The patches were extracted from an image from the Berkeley dataset [19], shown in Figure 2, right. Using a step size of 3, from the image were extracted 15,912 overlapping patches of size 15×15. Thus, the data have 15,912 observations, 225 inputs and 225 outputs. The outputs are the original image patches, and the inputs are the same image patches corrupted with Gaussian noise with standard deviation σ=10.

For all datasets, one hundred random splits of the data into an 80% train set and 20% test set were generated.

A two-layer NN with *h* hidden nodes and ReLU or Leaky ReLU activation was trained on the training parts of the random splits using different training algorithms and then tested on the corresponding test sets. The number of hidden nodes was h=64, except for the DAE data, where it was h=32 to obtain a low dimensional representation.

### 2.5. Methods Compared

The proposed ANMIN algorithm was compared with the stochastic gradient descent algorithm, the Adam optimizer [1] and the LBFGS algorithm [15]. The main focus is the SGD and Adam because the LBFGS blows up to values as large as $10^{26}$ many times during optimization, as one can see in Figure 3, left, for the abalone dataset, where 100 training losses are plotted. For this reason, for the LBFGS algorithm, for each run, the model that obtained the smallest training MSE was extracted and used for testing. The same approach was used for ANMIN (lines 17–20 of Algorithm 1).

ANMIN and Adam were experimented with two activation functions: ReLU (α=0) and leaky ReLU (α=0.1), while LBFGS and SGD were experimented only with ReLU. The methods with leaky ReLU are called ANMIN-L and Adam-L in experiments.

### 2.6. Implementation Details

All algorithms were implemented in PyTorch. The experiments were conducted on a Core I7 computer with 32 Gb RAM and GeForce 3080 GPU.

The code is available at https://github.com/barbua/ANMIN/ (accessed on 1 March 2023).

The SGD and Adam were trained for 300 epochs. The initial learning rate for Adam was 0.03, while for SGD, it was tuned for each dataset individually to avoid the loss blowing up and to obtain a small final loss value. The learning rate was decreased by a factor of 10 after every 100 epochs. The ANMIN algorithm was run for 30 iterations. The LBFGS was run for 60 iterations with a learning rate of 0.01.

For Adam and SGD, a batch size of 256 was used except for Year Prediction MSD, where a batch of 2048 was used. For LBFGS, a batch size of 100,000 was used. For ANMIN, a batch size of 256 was used. The batch size is used in the ANMIN to accumulate the matrices $U_{jl}$ from Equation (Equation 6), so different batch sizes result in identical solutions, and we have observed that the computation time is approximately the same for batch sizes between 128 and 2048. The value of the shrinkage parameter λ for ANMIN was λ=0.001.

## 3. Results and Discussion

Experiments are conducted on multiple real datasets to obtain insight about when the ANMIN method works well compared to regular gradient-based training and when it does not. We will see that the advantages of ANMIN are more clear when the input dimension is small relative to the number of hidden neurons. Then, simulations will be conducted on a nonlinear dataset where the input dimension and the number of observations can be modified as desired.

### 3.1. Real Data Experiments

The results as training, test MSEs and training and test $R^{2}$ are collected in Table 3. The best result and the results that obtained a *p*-value of p>0.01 based on a paired *t*-test significance comparison with the best results are shown in bold.

Figure 4, Figure 5 and Figure 6 show the training MSE loss and average $R^{2}$ curves for NNs with ReLU activation trained with the ANMIN method vs. Adam, SGD and LBFGS methods, respectively, on all 100 random splits. Additionally are shown the mean test MSE and $R^{2}$ curves with their standard deviation. The test MSEs and $R^{2}$ for ANMIN and LBFGS are based on the model with the smallest training MSE loss.

Looking at the MSE loss values in Figure 4, Figure 5 and Figure 6 and Table 3, one can see that the ANMIN method obtains significantly lower train loss values than Adam, SGD and LBFGS on four out of the five datasets. It obtains significantly lower test MSEs than Adam and LBFGS on four datasets and than SGD on three datasets. Moreover, it is never significantly outperformed on the test set by either SGD, Adam or LBFGS on any of the five datasets. The LBFGS algorithm performs very well only on abalone and is decent on Year Prediction MSD. It performs very poorly on the other three datasets.

Looking at the $R^{2}$ values from Figure 4, Figure 5 and Figure 6 and Table 3, we can see that the ANMIN method obtains significantly higher train $R^{2}$ values than SGD, Adam and LBFGS on four out of the five datasets. It obtains significantly higher test $R^{2}$ than Adam or LBFGS on four datasets and than SGD on three datasets. ANMIN is significantly outperformed by Adam and SGD on one dataset.

What do the datasets where the ANMIN method outperforms Adam and SGD have in common? One can notice that the one-output datasets in question all have low dimensional inputs, d≤13. Thus, the real data experiments seem to support the hypothesis that ANMIN has an advantage over standard gradient-based optimization when the input dimension of the data (hence the number of parameters in the hidden layer) is small. This hypothesis will be further explored in the simulations in Section 3.2. In that case, it seems that SGD and Adam have a hard time exploring the low dimensional parameter space and finding a deeper local optimum. Indeed, on the smallest dimensional dataset, which is the SDF decoder, ANMIN’s MSE was about three times smaller than SGD’s and Adam’s. On abalone and bike-sharing, the other two datasets with small numbers of features, ANMIN again significantly outperformed Adam and SGD on the training set, but SGD somehow generalized better on the test set for abalone.

The LBFGS algorithm did not do well on all datasets except for abalone, and in almost all cases, its training MSE values were so high that it was the worst of all methods evaluated.

The computation times for the methods evaluated are shown in Table 4. Comparing computation times, one can see from Figure 4 that a train loss value as large as the final Adam loss is obtained by ANMIN in 5–10 times less time on the three low-dimensional datasets, and more iterations bring even more improvements to the loss values. Even the total computation times (30 ANMIN iterations vs. 300 Adam) show a computation advantage for the ANMIN method, in the range of two times faster for the DAE to more than seven times for the three low-dimensional datasets.

These experiments indicate that the ANMIN method might have an advantage in some cases, e.g., when the input dimension *d* is small. This hypothesis is further investigated in the next section.

### 3.2. Simulations

In this section, simulations will be conducted to test the hypothesis that the advantage of ANMIN vs. Adam is more clear when the input dimension is small.

The simulated data has $x\in R^{d}$ sampled from $N(0,I_{d})$ and $y=\sin(∥x{∥}_{2}^{2})$. The dimension *d* was taken from d∈{1,3,10,30}. A number of N∈{1000,10,000} observations were sampled for training and an equal number for testing.

The model is a two-layer NN with h=64 hidden nodes and ReLU activation. It is worth noting that the model does not fit the data well when *d* is large.

The model was trained with 30 iterations of ANMIN (Algorithm 1). For comparison, it was also trained with 300 iterations of the Adam optimizer [1], starting with learning rate η=0.03, which was reduced by 10 every 100 iterations. The batch size was 256.

The plots of the MSE vs. iteration for 100 consecutive runs for different combinations of *n* and *d* are shown in Figure 7. For a better visualization, Figure 8 plots the train and test MSEs obtained by the two methods over the 100 runs, where the runs are sorted by decreasing test MSE for each method independently. This type of display can allow one to observe any trend between the train and test MSEs for each method. It would also allow one to obtain a better idea of the distribution of the test MSEs.

One can see that for small d∈{1,3}, the ANMIN can obtain both train and test loss values smaller than Adam. For d∈{10,30}, Adam obtains a smaller training loss, but it overfits more than ANMIN. In all situations, the ANMIN method obtains smaller test MSE values.

### 3.3. Ablation Study

This section investigates how the design decisions made for Algorithm 1 affect its performance. The design decisions that will be investigated are:Whether to use the minimum loss value encountered during the iterations instead of the final loss of the algorithm.When the linear system (Equation 9) is degenerate, whether to stop, reinitialize A randomly or continue with a pseudoinverse.

The ablation experiments are conducted on the bike-sharing dataset, which exhibits smaller log determinant values and therefore is more difficult to optimize. The results are summarized in Table 5.

In Experiment 1, the ANMIN algorithm is stopped the first time when the log determinant is less than −10,000 and the final loss is reported.

In Experiment 2, at each iteration when the log determinant is less than −10,000, the matrix A is re-initialized with random values. The model with minimum train loss over all iterations is reported with its training and test loss values.

In Experiment 3, the algorithm is run as described in Algorithm 1, except that the final model is reported instead of the model with minimum loss over the iterations.

Experiment 4 is exactly as described in Algorithm 1.

From Table 5, one can see that both the minimum over the iterations and the use of a pseudoinverse instead of a random restart help in obtaining a small final loss.

Figure 9 shows the MSE training loss values for the SDF decoder and bike-sharing datasets obtained at each iteration (left) and the minimum MSE obtained so far (right). One can see that the MSE loss at each iteration sometimes jumps (usually when the log determinant is small or −∞). However, the jumps are much smaller than those of the LBFGS method. This is because after each update of the matrix A, which can result in jumps, the matrix B is also updated to minimize the MSE.

### 3.4. Diagnosing the ANMIN Algorithm

As opposed to gradient-based algorithms, the ANMIN method offers some capabilities to diagnose the energy landscape during the optimization. For that, the condition number of $M+\lambda I_{(d+1)h}$ or the log determinant $\ln|M+\lambda I_{(d+1)h}|$ can be used to see how degenerate the local optimum that the ANMIN reaches is when fitting the hidden nodes A. Figure 10 and Figure 11 show the log determinant values $\ln|M+\lambda I_{(d+1)h}|$ encountered during optimization for the 100 ANMIN runs on the random splits for the ReLU and leaky ReLU activations respectively. Some curves are broken because the log determinant is −∞ at some iterations. One can see that for the one-output datasets, the log determinant has a tendency to decrease, while for the multiple-output DAE problem, the log determinant increases with the iterations. For the low-dimensional datasets (abalone and bike-sharing) the log determinant reaches a very small value, which makes the matrix $M+\lambda I_{(d+1)h}$ close to singular and the solution of Equation (Equation 9) less stable.

## 4. Conclusions

This paper introduced a method for training a two-layer NN with ReLU or leaky ReLU-like activation using the square loss by alternatively fitting the coefficients of each layer, while keeping the other layer and the activation pattern of the neurons fixed.

Experiments indicate that the method can obtain in some cases (e.g., when the input dimension is small) strong minima of the loss function that go beyond the capabilities of those obtained by gradient descent optimization. It can do so with a relatively small number of iterations (10–30), which means that it can work with large datasets that do not fit in the computer’s memory.

This work is experimental in nature, in that we introduce an algorithm and we observe that it works well experimentally, but we provide no proofs or theoretical guarantees that it will always work under certain assumptions.

This work can usher the possibility of using shallow NNs for many computation-sensitive applications by providing better optimization capabilities for shallow NNs that go beyond those of gradient descent algorithms.

It is known that tree ensembles, such as boosted trees and random forests, can be mapped to two-layer neural networks [20,21]. In this respect, this work can potentially enable better algorithms for training such deep ensembles by loss minimization, which are better at minimizing the loss and more computationally efficient.

However, the method has certain limitations since it involves the inversion of a matrix of size (d+1)h×(d+1)h. This can limit the applicability of the method when the input dimension and the number of neurons are both large (e.g., dh> 30,000). This drawback can be addressed by sparse neuron connections, where not all neurons are connected to all inputs, or by large-scale implementations that use multiple GPUs.

In the future, we plan to apply the ANMIN method to shape analysis using NNs and to large-scale object detection applications.

## Figures and Tables

**Figure 1 sensors-23-04072-f001:**
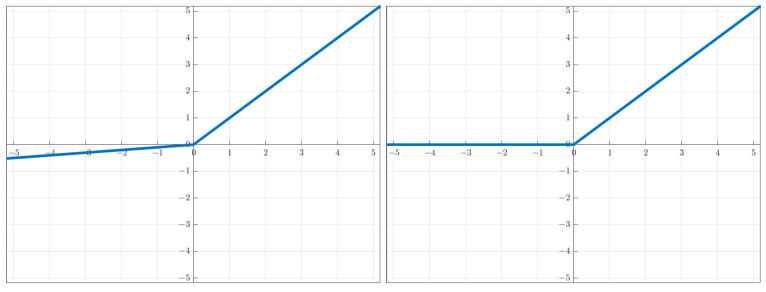
This work focuses on two-layer neural networks (NNs) with ReLU (rectified linear unit) and ReLU-like activation functions σ(x)=αx+(1−α)max(0,x), with α∈[0,1). Shown are the leaky ReLU (α=0.1, **left**) and the ReLU (α=0, **right**).

**Figure 2 sensors-23-04072-f002:**
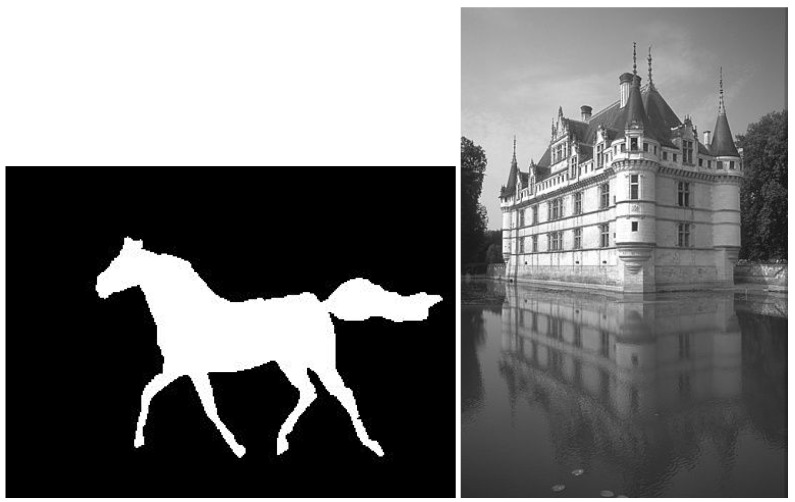
(**Left**): the binary shape image whose signed distance transform was used to train a shape decoder. (**Right**): the image used for training a denoising autoencoder (DAE).

**Figure 3 sensors-23-04072-f003:**
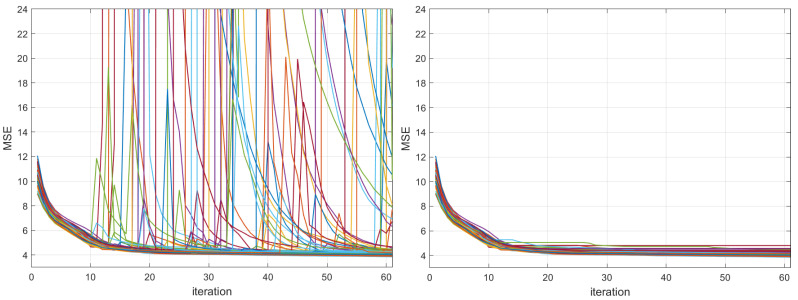
Training MSEs for LBFGS (limited memory Broyden–Fletcher–Goldfarb–Shanno algorithm) on 100 random splits of the abalone dataset, with the different random splits shown in different colors. (**Left**): the training MSEs have many places where LBFGS blows up. (**Right**): considering the minimum train MSE obtained so far at each iteration alleviates the problem.

**Figure 4 sensors-23-04072-f004:**
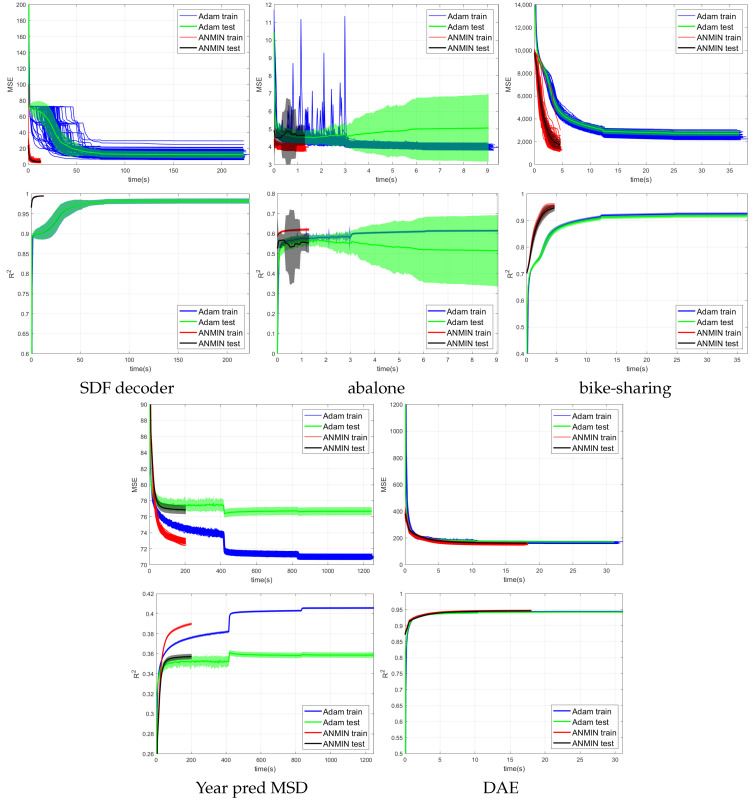
MSE and average $R^{2}$ vs. time (seconds) of 100 runs of training the NN with ReLU activation using the Adam, and ANMIN optimizers. Additionally plotted are the mean test MSEs and $R^{2}$ with standard deviation.

**Figure 5 sensors-23-04072-f005:**
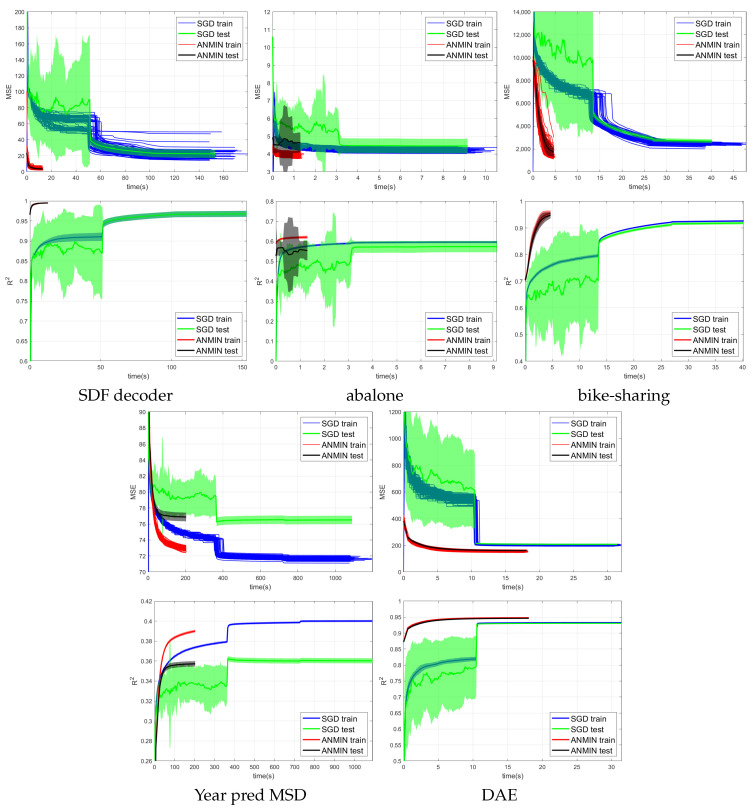
MSE and average $R^{2}$ vs. time (seconds) of 100 runs of training the NN with ReLU activation using the SGD and ANMIN optimizers. Additionally plotted are the mean test MSEs and $R^{2}$ with standard deviation.

**Figure 6 sensors-23-04072-f006:**
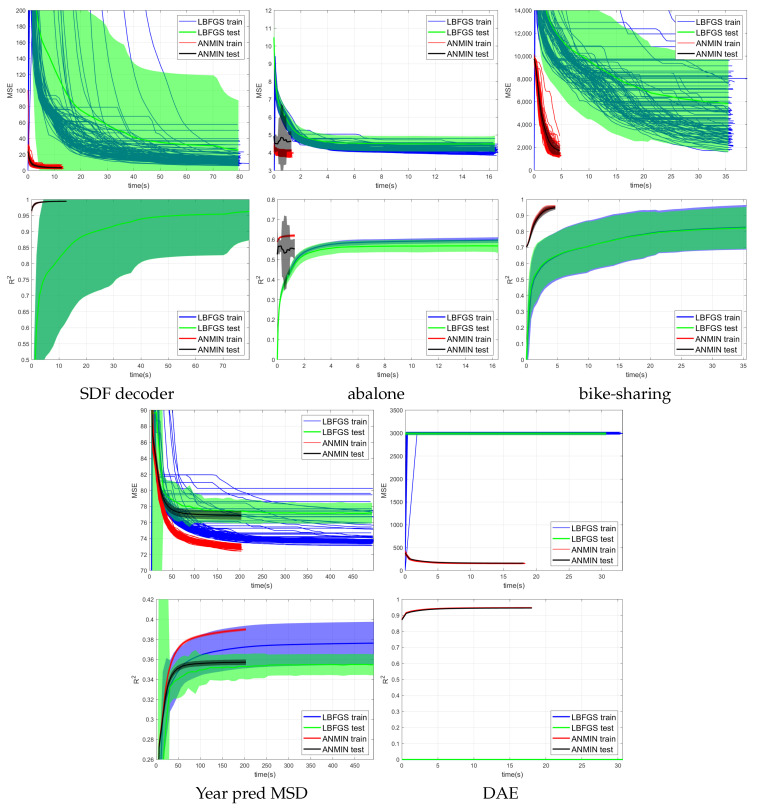
MSE and average $R^{2}$ vs. time (seconds) of 100 runs of training the NN with ReLU activation using the LBFGS and ANMIN optimizers. Additionally plotted are the mean test MSEs and $R^{2}$ with standard deviation.

**Figure 7 sensors-23-04072-f007:**
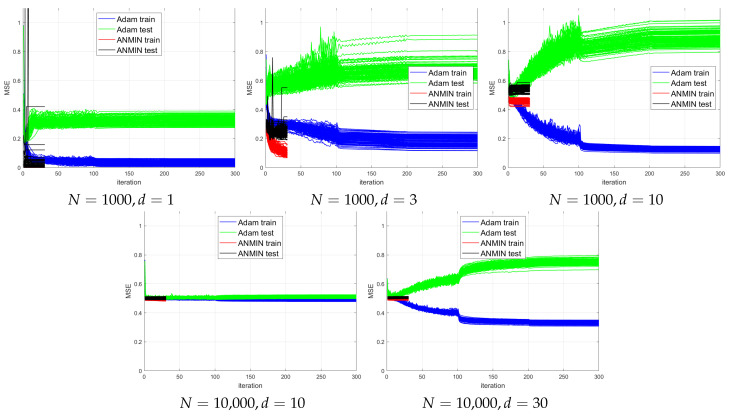
Simulation experiments. MSEs of 100 runs of training a 64 hidden node NN using 300 epochs of the Adam optimizer or 30 iterations of ANMIN (Algorithm 1).

**Figure 8 sensors-23-04072-f008:**
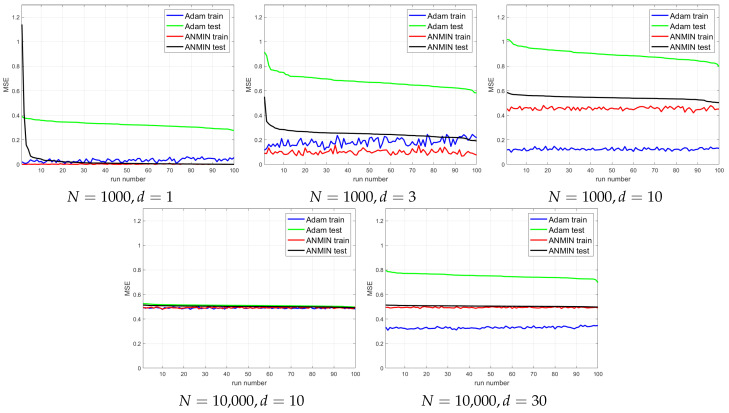
Simulation experiments. Final train and test MSEs obtained by the Adam and ANMIN over the 100 runs, sorted in decreasing order of the test MSE.

**Figure 9 sensors-23-04072-f009:**
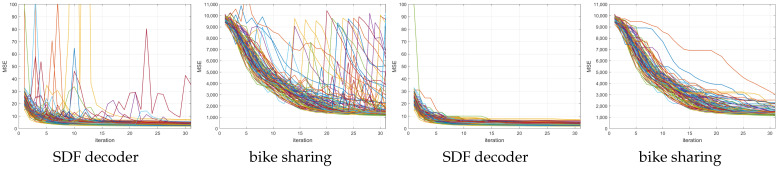
MSE plots vs. time (seconds) of 100 runs of training a 64 hidden-node NN using 30 iterations of ANMIN (Algorithm 1), with the different runs shown in different colors. Left two plots: the actual MSE values encountered at each iteration. Right two plots: the minimum MSE obtained so far at each iteration.

**Figure 10 sensors-23-04072-f010:**
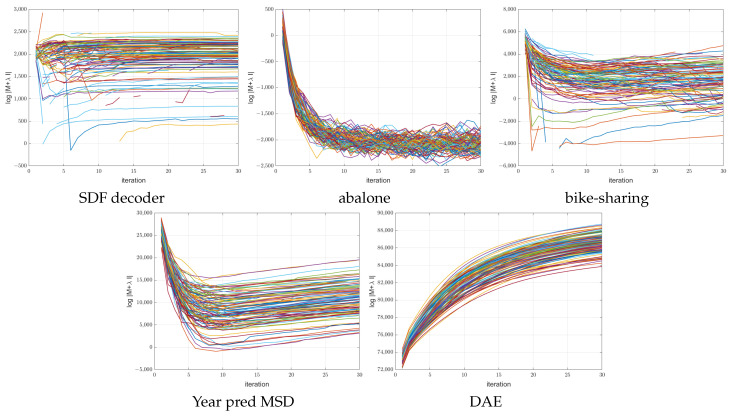
Plot of the $\ln|M+\lambda I_{(d+1)h}|$ vs iteration number for the 100 ANMIN (Algorithm 1) runs for NN with ReLU activation. The different runs are shown in different colors.

**Figure 11 sensors-23-04072-f011:**
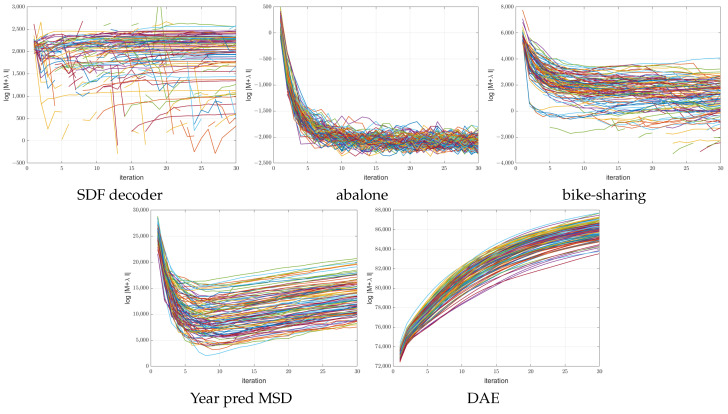
Plot of the $\ln|M+\lambda I_{(d+1)h}|$ vs. iteration number for the 100 ANMIN (Algorithm 1) runs for NN with leaky ReLU activation. The different runs are shown in different colors.

**Table 1 sensors-23-04072-t001:** Overview of related works with time complexity and theoretical global optimum guarantees.

	Complexity in			Loss	Global Optimum
Method	*N*	*d*	*h*	fn.	Guarantee
Adam [1]	*N*	*d*	*h*	any	No
SGD [4]	*N*	*d*	*h*	$\ell_{2}$	Yes (sigmoid and tanh activation)
ICA based [10,11]	poly(*N*)	poly(*d*)	exp(h)	some	No
Convex duality [8]	$N^{d}$	exp(d)	-	$\ell_{2}$	Yes
TD based [12]	poly(*N*)	poly(*d*)	poly(*h*)	-	No
Proposed	*N*	$d^{3}$	$h^{3}$	$\ell_{2}$	No

**Table 2 sensors-23-04072-t002:** The real datasets used in the experiments with their number of observations *n* and feature dimension *d*. Also shown is the number of NN hidden nodes *h* used in experiments.

Dataset	*N*	*d*	*h*
SDF decoder	73,080	2	64
Abalone	4177	7	64
Bike-sharing	17,379	13	64
Yearpred MSD	515,345	90	64
DAE	15,912	225	32

**Table 3 sensors-23-04072-t003:** Real data experiments. Average final train and test MSE values and $R^{2}$ (all with standard deviation (std) on five datasets. All results are obtained from 100 random splits into 80% train and 20% test. The best results and the ones not significantly worse (*p* > 0.01) are shown in bold.

	**Train MSE**					
Dataset	SGD	Adam	Adam-L	LBFGS	ANMIN	ANMIN-L
SDF decoder	22.43 (5.29)	12.35 (4.48)	9.46 (2.49)	25.71 (62.08)	**3.53** (1.05)	**3.45** (1.15)
Abalone	4.20 (0.08)	4.00 (0.09)	3.99 (0.09)	4.17 (0.16)	**3.94** (0.10)	**3.95** (0.09)
Bike-sharing	2422 (100)	2465 (181)	2477 (193)	5710 (4581)	**1457** (319)	**1555** (465)
YearPredMSD	**71.67** (0.16)	**70.98** (0.15)	**71.01** (0.14)	74.51 (2.55)	72.87 (0.18)	73.08 (0.21)
DAE	200.8 (3.1)	165.5 (4.9)	163.4 (4.5)	2990 (9.2)	155.2 (4.0)	**150.9** (2.7)
	**Test MSE**					
Dataset	SGD	Adam	Adam-L	LBFGS	ANMIN	ANMIN-L
SDF decoder	22.36 (5.18)	12.36 (4.46)	9.48 (2.49)	25.65 (61.96)	**3.56** (1.06)	**3.48** (1.14)
Abalone	**4.45** (0.41)	5.05 (1.89)	4.99 (1.71)	**4.50** (0.46)	**4.64** (0.55)	**4.66** (0.95)
Bike-sharing	2681 (171)	2721 (224)	2737 (239)	5850 (4280)	**1714** (385)	**1808** (493)
YearPredMSD	**76.51** (0.53)	**76.71** (0.51)	**76.68** (0.55)	77.17 (1.29)	**76.88** (0.54)	**76.82** (0.49)
DAE	207.1 (6.1)	170.8 (7.1)	168.7 (6.1)	2988 (36.8)	160.4 (5.6)	**156.1** (4.8)
	**Train *R*^2^**					
Dataset	SGD	Adam	Adam-L	LBFGS	ANMIN	ANMIN-L
SDF decoder	0.968 (0.008)	0.982 (0.007)	0.986 (0.004)	0.963 (0.089)	**0.995** (0.002)	**0.995** (0.002)
Abalone	0.596 (0.005)	0.615 (0.005)	0.615 (0.006)	0.598 (0.014)	**0.621** (0.007)	**0.620** (0.007)
Bike-sharing	0.926 (0.003)	0.925 (0.006)	0.925 (0.006)	0.826 (0.139)	**0.956** (0.010)	**0.953** (0.014)
YearPredMSD	0.400 (0.001)	**0.406** (0.001)	**0.406** (0.001)	0.376 (0.021)	0.390 (0.001)	0.388 (0.002)
DAE	0.933 (0.001)	0.945 (0.002)	0.945 (0.002)	0.000 (0.000)	0.948 (0.001)	**0.950** (0.002)
	**Test *R*^2^**					
Dataset	SGD	Adam	Adam-L	LBFGS	ANMIN	ANMIN-L
SDF decoder	0.968 (0.008)	0.982 (0.006)	0.986 (0.004)	0.963 (0.090)	**0.995** (0.002)	**0.995** (0.002)
Abalone	**0.573** (0.028)	0.515 (0.178)	0.521 (0.164)	**0.568** (0.035)	**0.555** (0.049)	**0.554** (0.079)
Bike-sharing	0.919 (0.005)	0.918 (0.006)	0.917 (0.007)	0.822 (0.130)	**0.948** (0.012)	**0.945** (0.015)
YearPredMSD	**0.360** (0.003)	0.359 (0.003)	0.359 (0.003)	0.355 (0.011)	0.357 (0.003)	0.358 (0.003)
DAE	0.931 (0.002)	0.943 (0.002)	0.944 (0.002)	0.000 (0.000)	0.946 (0.002)	**0.948** (0.002)

**Table 4 sensors-23-04072-t004:** Computation times (seconds) for the experiments from Table 3. Smallest times are shown in bold.

Dataset	SGD	Adam	Adam-L	LBFGS	ANMIN	ANMIN-L
SDF decoder	153	223	224	79.6	**12.8**	19.3
Abalone	9.1	9.1	9.1	16.4	1.3	**0.9**
Bike-sharing	40.2	36.7	36.0	35.5	4.7	**3.2**
Yearpred MSD	1089	1246	1084	493	**203**	208
DAE	31.4	31.1	31.4	30.7	**18.1**	**18.1**

**Table 5 sensors-23-04072-t005:** Ablation experiments. Average MSEs (with std) obtained on the bike-sharing dataset by the ANMIN Algorithm 1 with different parts removed.

Experiment	min	pinv	Train MSE	Test MSE
1	−	−	8919 (4095)	9010 (4056)
2	+	−	1826 (660)	2092 (662)
3	−	+	2199 (1972)	2450 (1955)
4 (Algorithm 1)	+	+	1457 (319)	1714 (385)

## Data Availability

The following publicly available datasets have been used in experiments: Abalone, Bike-sharing and Year Prediction MSD from the UC Irvine Machine Learning Repository [16], the Weizmann horse dataset [18] and the Berkeley dataset [19]. The DAE and SDF decoder data can be found at: https://github.com/barbua/ANMIN/tree/main/data (accessed on 17 April 2023).

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
