# Peer review of "Training a Two-Layer ReLU Network Analytically"

_sensors, 2023, doi:10.3390/s23084072_

Round 1

Reviewer 1 Report

The manuscript written by author has been qualified for publicatio after minor corrections

1. Improve english

2. Increase and update refrences

3. Add more related work

4. Add organizaton part in the last of introduction section

5. Improve formatting

6. Add improvement analysis.

Kindly perform above corrections and resubmit again for publication. 

Author Response

We thank the reviewer for the valuable feedback.

R1: The manuscript written by author has been qualified for publication after minor corrections

  1. Improve English

Reply: The paper was proofread again.

  1. Increase and update references

Reply: We have added more references, mainly to the Related Work section.

  1. Add more related work

Reply: We have added a few more papers to related work, but there are very few related works in the literature.

  1. Add organization part in the last of introduction section

Reply: We have added a paragraph at the end of introduction giving an overview of the paper organization.

  1. Improve formatting

Reply: We have reorganized the figures and made them larger to be more legible.

  1. Add improvement analysis.

Reply: The analysis of the improvement over the existing Adam, SGD and LBFGS methods can be found in Section 3.1

Reviewer 2 Report

This paper explored an algorithm for training two-layer neural networks with ReLU-like activation and the square loss. In general, this paper is well presented, and the experiment is also relatively sufficient. Therefore, I think this paper is suitable for publication in the current format.

Author Response

We thank Reviewer 2 for the vote of confidence.

Reviewer 3 Report

For shallow Relu NN, the output layer weights and bias are linear parameters when the hidden layer is fixed, while the hidden layer weights and biases are nonlinear. 

It is nice to separate these two types of parameters, especially for the output layer parameters, solving the linear system in (4) corresponds the analytic critical point.

For the nonlinear parameters A (hidden layer weights), the proposed method solve for it by fixing the firing pattern of the neurons and reduce the loss function to a convex problem, but this is by no means the the analytic critical point corresponds \partialL/partialB=0. As an approximation method,  since the matrix G is a function of B, fixed G to optimize L wrt B is only valid when the update of B cause minor changes of G, which in most cases not valid.

Without theoretic analysis on the assumption of fixing G (and M) to iterate back and forth A and B, I am not convinced conceptually this leads to a stable algorithm.

For the algorithm itself, how the two linear systems are solved? direct matrix inversion or other iterative method?, if the former, are the two matrices always invertible?

The conclusion of the paper is a little vague: "Experiments indicate that the method can obtain in some cases strong minima...". 

Some minor comments:

(1) matrix U_ij in (6): dimension is (d+1)x(d+1)?

(2) Matrix M in (7): dimension should (d+1)hx(d+1)h?

(3) table 2: what is ANMIN-L?

Author Response

We thank Reviewer 3 for the valuable feedback.

R3: For shallow Relu NN, the output layer weights and bias are linear parameters when the hidden layer is fixed, while the hidden layer weights and biases are nonlinear. 

It is nice to separate these two types of parameters, especially for the output layer parameters, solving the linear system in (4) corresponds the analytic critical point.

For the nonlinear parameters A (hidden layer weights), the proposed method solve for it by fixing the firing pattern of the neurons and reduce the loss function to a convex problem, but this is by no means the the analytic critical point corresponds \partialL/partialB=0. As an approximation method,  since the matrix G is a function of B, fixed G to optimize L wrt B is only valid when the update of B cause minor changes of G, which in most cases not valid.

Without theoretic analysis on the assumption of fixing G (and M) to iterate back and forth A and B, I am not convinced conceptually this leads to a stable algorithm.

Reply: This paper is of an experimental nature, where we report to the community experiments based on the proposed algorithm and observed that the algorithm works well on all five datasets evaluated. We leave the proof of why it works to other researchers that are more skilled in theoretical guarantees for neural networks. One could think of this work as a conjecture in mathematics or an experimental paper in physics.

R3: For the algorithm itself, how the two linear systems are solved? Direct matrix inversion or other iterative method? If the former, are the two matrices always invertible?

Reply: The linear systems are solved using a more numerically stable version of matrix inversion. The matrix in Eq. (4) is always invertible due to the parameter . The matrix from Eq. (9) is not always invertible, which is why we check for the log determinant and use a pseudo-inverse when it is not invertible.

R3: The conclusion of the paper is a little vague: "Experiments indicate that the method can obtain in some cases strong minima...". 

Reply: We clarified that the cases in question are when the input dimension is small.

Some minor comments:

(1) matrix U_ij in (6): dimension is (d+1)x(d+1)?

Reply: Yes, fixed.

(2) Matrix M in (7): dimension should (d+1)hx(d+1)h?

Reply: Yes, fixed.

(3) table 2: what is ANMIN-L?

Reply: ANMIN with leaky ReLU activation. We added a sentence in Section 2.5 to clarify this.

Reviewer 4 Report

Dear authors, this is exciting work on training neural networks analytically. Neural networks are generally trained with variants of the gradient descent algorithm. However, this work introduces a method for training a two-layer NN with leaky ReLU-like activations and the square loss analytically by solving ordinary least squares equations. The work has archival value, but this review has some concerns that must be addressed before it is considered for publication in this respected journal.

1) The abstract does not sell the paper properly. It should contain values obtained from comparisons with other works. Please, let the prospective reader know how good your proposal is compared to the state-of-the-art.

2) There is no related work section. The authors should add such a section and explain clearly what the differences are between the prior work and the solution presented in this paper.

3) The number of cited works is very small. The literature review is incomplete. Several relevant references are missing.

4) The authors should add a table that compares the key characteristics of prior work to highlight their differences and limitations. The authors may also consider adding a line in the table to describe the proposed solution.

5) Figures 4 onwards are too small. Consider splitting them into separate figures.

6) The paper has some typos and grammar errors. Authors need to proofread the paper to eliminate all of them.

7) To ensure the reproducibility of the results, the code of the proposed solution should be made public on a website.

8) The experiments have been carried out with some data that is not publicly available. Authors should make it available on some public websites so that experiments can be reproduced.

Author Response

We thank Reviewer 3 for the valuable feedback.

R4: Dear authors, this is exciting work on training neural networks analytically. Neural networks are generally trained with variants of the gradient descent algorithm. However, this work introduces a method for training a two-layer NN with leaky ReLU-like activations and the square loss analytically by solving ordinary least squares equations. The work has archival value, but this review has some concerns that must be addressed before it is considered for publication in this respected journal.

1) The abstract does not sell the paper properly. It should contain values obtained from comparisons with other works. Please, let the prospective reader know how good your proposal is compared to the state-of-the-art.

Reply: We have added text to clarify that the proposed method obtained significantly smaller training loss values than SGD and Adam on four out of the five datasets evaluated.

2) There is no related work section. The authors should add such a section and explain clearly what the differences are between the prior work and the solution presented in this paper.

Reply: We added a Related Work section and moved the relevant material there. We also added text to clarify the differences between our work and each of the existing works.

3) The number of cited works is very small. The literature review is incomplete. Several relevant references are missing.

Reply: We have added a few more related works, but the literature on non-gradient based methods for neural networks is very limited and we had a hard time finding other works. It would be helpful if the Reviewer would point us to what relevant references we are missing and we will gladly add them.

4) The authors should add a table that compares the key characteristics of prior work to highlight their differences and limitations. The authors may also consider adding a line in the table to describe the proposed solution.

Reply: We have added a table comparing the computation complexity in N, d, h, the loss function that is optimized and the global optimum guarantees for different algorithms from the related works section. Some papers are just theoretical without an algorithm, so they were not included.

5) Figures 4 onwards are too small. Consider splitting them into separate figures.

Reply: We split the figures into many figures and made them larger.

6) The paper has some typos and grammar errors. Authors need to proofread the paper to eliminate all of them.

Reply: We have proofread the paper again.

7) To ensure the reproducibility of the results, the code of the proposed solution should be made public on a website.

Reply: We published the code on GitHub at https://github.com/barbua/ANMIN/, and added a link to it in the paper.

8) The experiments have been carried out with some data that is not publicly available. Authors should make it available on some public websites so that experiments can be reproduced.

Reply: We put the data together with the code on GitHub.